# Atomic Nanogenerators in Targeted Alpha Therapies: Curie’s Legacy in Modern Cancer Management

**DOI:** 10.3390/ph13040076

**Published:** 2020-04-23

**Authors:** Mareike Roscher, Gábor Bakos, Martina Benešová

**Affiliations:** 1German Cancer Research Center (DKFZ), Division of Radiooncology/Radiobiology, Im Neuenheimer Feld 280, 69120 Heidelberg, Germany; mareike.roscher@dkfz-heidelberg.de; 2German Cancer Research Center (DKFZ), Research Group Molecular Biology of Systemic Radiotherapy, Im Neuenheimer Feld 280, 69120 Heidelberg, Germany; g.bakos@dkfz-heidelberg.de

**Keywords:** atomic in vivo nanogenerators, α-emitters, thorium-227 (^227^Th), actinium-225 (^225^Ac), radium-223 (^223^Ra), coordination chemistry, nuclear recoil effect, radiopharmaceuticals, targeted alpha therapies (TAT)

## Abstract

Atomic in vivo nanogenerators such as actinium-225, thorium-227, and radium-223 are of increasing interest and importance in the treatment of patients with metastatic cancer diseases. This is due to their peculiar physical, chemical, and biological characteristics, leading to astonishing responses in otherwise resistant patients. Nevertheless, there are still a few obstacles and hurdles to be overcome that hamper the broader utilization in the clinical setting. Next to the limited supply and relatively high costs, the in vivo complex stability and the fate of the recoiling daughter radionuclides are substantial problems that need to be solved. In radiobiology, the mechanisms underlying treatment efficiency, possible resistance mechanisms, and late side effect occurrence are still far from being understood and need to be unraveled. In this review, the current knowledge on the scientific and clinical background of targeted alpha therapies is summarized. Furthermore, open issues and novel approaches with a focus on the future perspective are discussed. Once these are unraveled, targeted alpha therapies with atomic in vivo nanogenerators can be tailored to suit the needs of each patient when applying careful risk stratification and combination therapies. They have the potential to become one of the major treatment pillars in modern cancer management.

## 1. Introduction

“*A scientist in his laboratory is not a mere technician: he is also a child confronting natural phenomena that impress him as though they were fairy tales.*”—Marie Skłodowska Curie (07.11.1867–04.07.1934) [1].

With the discovery of polonium and radium more than a century ago (1898) for which she was awarded with her second Nobel Prize (1903 and 1911), Marie Skłodowska Curie laid the foundation for the development of nuclear medicine [2]. Its progress is also closely related to other important discoveries within many disciplines. These include but are not limited to the discovery of X-rays by Wilhelm Conrad Röntgen (1895), the phosphorescence of uranium salts by Henri Becquerel (1896) [3], the tracer principle by György Hevesy (1920) [4], and the invention of artificial radioactivity (Irène and Frédéric Joliot-Curie) [5], as well as of the cyclotron (Ernest Lawrence) [6], both in 1934. Finally, another important aspect was related to the availability of nuclear facilities, for example within the infamous Manhattan project [7]. However, already during Curie’s time, the historically first cancer therapies using radium (radium-226, ^226^Ra, τ_½_ = 1600 y) were introduced. These therapies were based on the knowledge about the physiological effects of radium and led to the first successful treatments using atomic in vivo nanogenerators, also preparing a solid basis for the concept of targeted alpha therapy (TAT) as we know it today [8,9,10,11].

Systemic endoradiotherapies using beta-minus- and alpha-emitters (β^–^ and α) are a rapidly expanding field in cancer treatment, and the use of mainly β^–^-particles is considered as a standard of care for thyroid cancer, bone metastases, and pain, as well as for non-Hodgkin lymphoma. Since the Food and Drug Administration (FDA) approval of radium-223 dichloride ([^223^Ra]RaCl_2_, Xofigo, Bayer) in 2013 [12], both the pre-clinical and clinical potential of α-particles—with a focus on the management of metastatic cancer diseases—has been fiercely investigated. The research is not restricted to cancer management alone but also covers the treatment of HIV as well as fungal and bacterial infections with promising results (for a review, see [13,14]). α-particles are generally considered to be highly promising due to their characteristics—the short particle range in tissue (40–100 µm or 2–10 cell diameters) and high-linear energy transfer (LET; ≈80 keV/µm)—which result in a high cytocidal effect for malignant tissue while sparing healthy surrounding tissue [15,16,17]. Nevertheless, only a handful of the many known α-emitters can be exploited in the clinical setting because of their inherent physical characteristics. Among these are the shorter-lived α-particle-emitting radionuclides bismuth-213 (^213^Bi, τ_½_ = 45.59 min), and astatine-211 (^211^At, τ_½_ = 7.2 h), as well as the atomic in vivo nanogenerators thorium-227 (^227^Th, τ_½_ = 18.7 d), actinium-225 (^225^Ac, τ_½_ = 9.9 d), and radium-224/-223 (^224/223^Ra, τ_½_ = 3.6 d and 11.4 d), which decay by several α-emissions [18,19,20,21].

For optimized therapeutic efficacy, TAT relies on stable targeting vectors specifically addressing malignant tissue and chelating moieties displaying stable in vivo behavior. According to the “magic bullet” (“Zauberkugel”) approach proposed by Paul Ehrlich (1900), the cytotoxic payload is delivered selectively to the malignant cells [22]. The vectors can be represented by small molecules, peptides, antibodies, or antibody derivatives, as well as by nanocarriers (as a full review of these would be exceeding the limits of this article, please refer, for instance, to [23,24,25,26,27,28]). [^223^Ra]RaCl_2_, however, represents an exception due to its intrinsic bone-targeting properties, mimicking the chemical behavior and metabolic pathway of calcium, as both elements belong to the alkali earth metals. This entity is, thus, applied in its un-chelated form and without a dedicated targeting vector [12,29,30].

Pre-clinical and clinical studies show a higher tumor response rate of TAT and the ability to overcome treatment resistance for endoradiotherapy with β^–^-emitters, as well as standard radiotherapy and chemotherapy. For example, among the patients suffering from metastatic castration-resistant prostate cancer (mCRPC), third-line treatment with a β^–^-emitting radiopharmaceutical [^177^Lu]Lu-prostate-specific membrane antigen (PSMA)-617 shows only 45% biochemical response rate, whereas 40% of these patients do not respond at all. On the other hand, treatment with an α-emitting analogue, [^225^Ac]Ac-PSMA-617, shows higher overall response rates [31,32,33,34]. Despite this seemingly favorable outcome, TAT tends to have more severe side effects such as the permanent deterioration of the salivary glands. Further still, some patients experience relapse regardless of the treatment with an α-emitting radiopharmaceutical [34]. Additional off-target effects caused by the nuclear recoil effect can lead to further toxicities and presumably also to late side effects [35].

In this review, the chemistry, physics, and biology of atomic in vivo nanogenerators are briefly highlighted and summarized. The main intention of the authors is to present a golden thread, connecting examples of the phenomenological aspects (physicochemical properties, chelation chemistry, nuclear recoil, and radiobiological facets), with the various scientific achievements and clinical successes made thus far. Light will be shed on the different obstacles and hurdles to be overcome and the open issues to be investigated in order to broaden the knowledge on TAT. Furthermore, emphasis is also given to different strategies to optimize the clinical acceptance and application of TAT in the future, which will hopefully lead to new individualized treatment schemes in modern cancer management.

## 2. Physicochemical Properties of ^225^Ac and ^227^Th

Actinium (from the Greek *actinos* = ray) was discovered by André-Louis Debierne (1899) in leftovers of uranium ore, which also enabled the discovery of radium and polonium by Marie Skłodowska Curie [36]. Actinium preferentially exists in the oxidation state +3, and has no electrons in its outermost shell (electron configuration 5*f*
^0^ 6*d*
^0^) [37]. ^225^Ac (Figure 1) appears to be the most promising actinium radioisotope for utilization in TAT [19,38]. Gamma spectroscopy—in the case of ^225^Ac—is challenging, as it only emits γ-rays with relatively low intensity (≈1%) [39,40,41]. Thus, its first daughter radionuclide (^221^Fr; τ_½_ = 4.8 min, E_γ_ = 218 keV, I = 11.4%) is commonly used for the characterization of ^225^Ac via gamma spectroscopy, as ^225^Ac forms a so-called *secular equilibrium* (τ_½, MOTHER >>>_ τ_½, DAUGHTER_) with its daughters. 

Thorium (named after the Scandinavian god of war, Thor) was discovered by Jōns Jacob Berzelius (1832) from the mineral rock thorite [42]. Despite the fact that thorium preferentially exists in the oxidation state +4, it can possess different coordination numbers determined by the concrete chelating ligand [43]. ^227^Th appears to be the most promising thorium radioisotope for utilization in TAT [21,44]. The decay scheme of ^227^Th is relatively similar to the one of ^225^Ac, however, the half-life is almost doubled (Figure 2). In contrast to ^225^Ac, ^227^Th possesses a gamma ray (E_γ_ = 235 keV, I = 12.9%) that can be easily detected by gamma spectroscopy. ^227^Th decays to ^223^Ra in so-called *transient equilibrium* (τ_½, MOTHER >_ τ_½, DAUGHTER_) and ^223^Ra decays to ^219^Rn (τ_½_ = 4.0 s) in a *secular equilibrium* (τ_½, MOTHER >>>_ τ_½, DAUGHTER_).

## 3. Coordination Chemistry

Proper chelating agents for the stable coordination of ^227^Th (and ^223^Ra) as well as of ^225^Ac are of utmost importance [45,46,47]. However, no single chelating agent can properly bind all daughter radionuclides over the entire decay chain. Overall, the stability of radiopharmaceuticals for TAT is based on many different characteristics, including (i) the coordination properties of the mother radionuclide (compatibility of the radionuclide with the chelating agent; Table 1 and Table 2), (ii) the coordination properties of the daughter radionuclide (compatibility of the radionuclide with the chelating agent; Table 1 and Table 2), (iii) the kinetics of complexation (time, temperature, pH), (iv) the thermodynamic stability (stability of radionuclide-chelating agent complex in a solution), and (v) the kinetic inertness (competition with many ions and chelating agents in body fluids). Moreover, both the radiolytic effect (highly reactive radicals) and the recoil effect (energy imparted to the daughter nuclei) strongly influence the quality of the resulting radiopharmaceuticals [48,49]. From a thermodynamic stability point of view, macrocyclic chelators (rather than acyclic) could be more beneficial. Moreover, the more stable “in-cage” complexes (radiolabeling at elevated temperatures and/or by the use of microwave systems) could prove to be of high importance when atomic in vivo nanogenerators are applied. Radiolabeling temperature, thus, plays a special role in the in vivo complex stability, as it can determine if so-called “in cage” or “out-cage” macrocyclic complexes are formed [50]. From a biological point of view, a matching radiometal–chelate chemistry is central for the success of specific and targeted therapies. Radionuclides, which are not delivered to the target tissue (i.e., malignant tissue) but are distributed through the body and to non-target organs, can lead to toxicities and unwanted side effects, as well as to a reduced dose in the target tissue [17,51].

Despite ongoing research efforts, a thermodynamically stable and kinetically inert chelator for radium (II) complexation is still not available. This situation also hampers the broader utilization of the inherently bone-seeking ^223^Ra for additional TAT applications apart from the management of skeletal metastases. There are some attempts to develop suitable chelators, for example cage-like sulfonated calix[4]crowns [52]. However, the first daughter radionuclide ^219^Rn is a noble gas that represents a clear challenge when considering the overall complex stability. Other attempts focus on different inorganic nanomaterials that are known for their ability to attract and bind ^223^Ra via sorption mechanisms [53].

In the case of actinium (III), the knowledge about its coordination chemistry is hindered by the lack of a stable actinium isotope. The large ionic radius of Ac^3+^ (126 ppm), which is the largest of all +3 nuclides, gives rise to kinetically labile complexes. The use of the current state-of-the-art DOTA chelator (1,4,7,10-tetraazacyclododecane-1,4,7,10-tetraacetic acid, also known as tetraxetan; Figure 3) for ^225^Ac labeling is suboptimal (thermodynamic preference for smaller ions), but still feasible [31,34]. On the other hand, a very interesting finding of Khabibulin et al. [54] demonstrated the absence of any chemical bonding for the Fr–DOTA complex. This complex should be of utmost importance as ^221^Fr is the first daughter radionuclide of ^225^Ac and the overall stability of the radiopharmaceutical is, thus, not only affected by the recoil effect but also by the coordination incompatibility upon ^225^Ac decay. More recently, the bis-picolinate ligand Macropa (Figure 3) and its derivatives proved the ability to form stable complexes with large lanthanide ions, which is an essential feature for ^225^Ac coordination. The labeling with these ligands is usually performed at room temperature within a few minutes [55]. Such labeling conditions are of a high benefit when temperature-sensitive vectors (e.g., antibodies) are used. Moreover, fast complexation might also improve the general stability of radiopharmaceuticals while diminishing the impact of both the radiolytic as well as the recoil effect [56]. Finally, another benefit of short complexation times might also be represented by the lower concentration of daughter progenies in the labeling solution because these can successfully compete with the mother radionuclide for the donor atoms of the chelating agent.

Thorium (IV) has a relatively complicated coordination chemistry that is similar to that of zirconium (IV). At pH >7, thorium forms various water-insoluble oxides and, thus, precipitates out of a solution. Thorium is known as an oxophilic metal, which means it prefers oxygen donors for metal coordination [57]. Thus, the use of DOTA as chelator (with four coordinating nitrogen donors) is suboptimal, but still feasible. On the other hand, the Me-3,2-HOPO chelator (Figure 3) and its derivatives—with eight available oxygen donors for coordination and extremely high stability constant with thorium—appear to be more suitable chelating agents for ^227^Th coordination. The labeling with these agents is usually performed at room temperature within one hour [58]. Such labeling conditions are still favorable for temperature-sensitive vectors, but especially, from the clinical point of view, the labeling time is relatively long and might affect the general stability of some radiopharmaceuticals. Finally, on the basis of the fact that the chemistry of thorium mimics the chemistry of zirconium (oxidation state +4, most common coordination number 8), ^89^Zr (τ_½_ = 3.3 d), a widely used positron-emitting radionuclide, could be employed as a diagnostic match to therapeutic ^227^Th within the same pharmaceutical [59].

## 4. Nanogenerators and the Nuclear Recoil Effect

The concept of atomic in vivo nanogenerators such as ^225^Ac evolved to overcome the short half-life of clinically applied α-particle-emitting radionuclides such as the ^225^Ac daughter radionuclide ^213^Bi. Another important driving force for applying atomic in vivo nanogenerators is their higher energy deposition to tumor sites. These usually decay via a cascade of multiple α-emitting daughters (e.g., net α-energy of totally 28 MeV for ^225^Ac), resulting in higher levels of cytotoxic radiation in target tissues with less activity applied [39,60]. For example, ^225^Ac was shown to have the same therapeutic efficiency with approximately 1000 times lower activity than its daughter ^213^Bi alone [61]. ^225^Ac has a >300-fold longer half-life than ^213^Bi, and hence allows the use of targeting vectors with slower distribution kinetics [51]. Nevertheless, despite these encouraging successes in the selective destruction of tumor cells, efficient solutions for the highly variable chemistry of the daughter radionuclides and the nuclear recoil effect are still yet to be found.

The nuclear recoil effect is based on the conservation of momentum law occurring with the release of the α-particle. An approximate value of the recoil energy (*E_t_*) can be calculated from the rest mass of the α-particle (*m_α_*), the mass of the recoil daughter (*M_r_*), and the decay energy (*Q*) [50]:(1)$$E_{t}=\frac{m_{\alpha }}{M_{r}}\cdot Q$$

The recoil energy passed on to the daughter nuclide is around 100–200 keV—about 1000–10,000 times higher than any binding energy between the atoms in a chemical bond—hence, the daughter radionuclide is released from the targeting vector (Figure 4) [20,50].

Its redistribution depends on where the daughter radionuclide is released and on the distance it can cover due to the given recoil energy. Diffusion processes and active transport, such as in the blood, as well as the intrinsic affinity of the radionuclide for certain organs (Table 3) further have an impact on the (re)location [20].

As such, the half-lives of the mother and daughter radionuclides determine the time they can need to reach the target and also their possible toxicity to healthy organs [20,50] (Figure 5). In addition, biodistribution as well as pharmacokinetic and pharmacodynamic parameters such as the effective half-life of TAT pharmaceuticals also play an important role in the (re)location of the mother and daughter radionuclides.

Due to their inherent properties, chelator-free radionuclides can cause various side effects in different organs (Table 3). The dose-limiting side effect of ^227^Th is bone marrow suppression leading to a decrease in white blood cell count [75]. This side effect arises from the biodistribution of its daughter radionuclide ^223^Ra, which is well characterized [71,76]—it has a fast blood clearance and is either taken up and trapped on the bone surface (hydroxyapatite) or excreted via the intestines [69]. The daughter radionuclides of ^223^Ra itself most probably contribute to the radiation burden in the vicinity of the site of ^223^Ra decay, as they have half-lives of only milliseconds to minutes [75].

Non-complexed ^225^Ac accumulates in the liver and to a lesser extent in bone [62]. Its first daughter, ^221^Fr, accumulates in the kidney [63], whereas no significant data for the biodistribution of the daughter radionuclide ^217^At could be measured because of its short half-live (32 ms). It is likely that its decay, as proposed for the daughter nuclides of ^223^Ra, also occurs close to the decay site of ^225^Ac or in the circulation [77]. Non-complexed ^213^Bi with a half-life of 45 min accumulates in the renal cortex where it is most probably coordinated by metallothionein-like proteins [64], exerting unwanted toxic effects.

Three main approaches have been suggested and tested in order to limit the distribution of recoil daughter radionuclides [20]. In the first, the mobility of the daughter radionuclide is limited if the vector is taken up and internalized quickly in the target tissue [39]. The probability of the mother radionuclide to decay in the blood stream is reduced and the free diffusion/active transport of the daughter radionuclides is hampered. The direct administration of the radiopharmaceutical to the target site via injection and the use of nanoparticles encapsulating the radioactivity are the two other approaches. Especially for nanoparticles, it has been hypothesized that the daughter radionuclides are caught inside of the particle while the α-particles can freely deposit their dose in the target tissue ([78]; for an excellent review regarding the different forms of nanoparticles, please refer to Holzwarth et al. [27]). The in vivo proof of concept, however, has to face some challenges as well. If not as a targeted approach, that is, functionalization of the nanoparticles with tumor-addressing vectors, the nanoparticle biodistribution—due to the large size of the particles—depends on the enhanced permeability and retention (EPR) effect of tumors, in which the blood vessels develop abnormally and have a wide fenestration, leading to the accumulation of the nanoparticles [79]. The liver and spleen in particular also have a relatively wide vessel fenestration, and thus also here an unwanted accumulation of the particles might occur. For a targeted approach using antibody-targeted nanoparticles in a murine model, a high specific lung uptake (i.e., antigen expressing tissue) could be observed [80]. However, a comparable high uptake of the nanoparticles in liver and spleen was detected irrespective of an eventual targeting with the antibodies. Furthermore, the main hepatic excretion of the particles can additionally lead to a high accumulation of the α-emitters, resulting in damage to the organ [78].

The accumulation and excretion of the parent and daughter radionuclides itself can be altered. Different strategies show potential to reduce the toxicities to dose-limiting organs and to further improve TAT outcome and safety for the treated patients. The daughter nuclides of ^225^Ac, ^213^Bi, and most likely also ^221^Fr, accumulate in the kidney where they can lead to nephropathy [63]. In terms of kidney protection, different strategies such as accelerated kidney excretion and prevention of renal absorption have been tested. As ^213^Bi binds to metallothionein-like proteins in the renal tubular cells, dithionein/metal chelators such as DMSA (meso-2,3-dimercaptosuccinic acid), DMPS (2,3-dimercapto-1-propanesulfonic acid), and Ca-DTPA (calcium-diethylenetriamine pentaacetate) can scavenge non-complexed ^213^Bi and lead to a faster excretion, with DMPS being the most efficacious one [63,81,82,83,84,85]. However, this strategy also increases the circulating ^213^Bi in the blood, with thus far unknown long-term effects. Diuretics such as furosemide and chlorothiazide have been shown to prevent the tubular re-absorption of ^221^Fr and hence also lead to a decrease of ^213^Bi accumulation in the analyzed murine model. Combining diuretics and chelators could even further decrease the renal ^213^Bi accumulation in this study [63]. As well as this, the co-injection of non-radioactive bismuth citrate can block the renal uptake of radioactive ^213^Bi due to competition for binding sites [63]. Depending on the applied TAT pharmaceutical, such as for peptide receptor radionuclide therapy in general, the kidney might be additionally protected via amino acids such as l-lysine. In a murine model of neuroendocrine tumors, the renal re-absorption of [^213^Bi]Bi-DOTATATE increased the therapy’s safety and efficacy [86]. For ^223^Ra, oral administration of BaSO_4_ can reduce its uptake in the large intestine as a co-precipitating agent [87].

## 5. Radiobiological Considerations Using α- and β^–^-Emitters for Endoradiotherapy

In radiobiology, three key factors are generally involved in the response of a tumor cell to radiotherapy: proliferation, hypoxia, and cellular radiosensitivity, which itself is determined by DNA repair, cell growth characteristics, and genome instability [88]. Playing a key role in a successful tumor treatment, these factors have to be considered carefully when choosing the radiation source for radiotherapy. As radiation-induced biological effects in cells and the relevant signaling pathways are a wide field on their own, it is outside of the scope of this article to discuss them in more detail. However, for a more detailed overview on the biological responses triggered after (endo)radiotherapy, we would like to refer the interested reader to the excellent reviews of Pouget et al. [89,90]. Instead, we rather focus here on the comparison between α- and β^–^-emitters and their properties, which need to be considered when planning the radiation treatment and the applied source for endoradiotherapy.

The therapeutic efficacy of endoradiotherapy can be maximized by matching the physical properties of the radionuclide (e.g., decay mode, half-life, effective range, relative biological effectiveness (RBE)) to the respective tumor entity (e.g., tumor mass, size, radiosensitivity, heterogeneity, or metastases). β^–^-particle-emitting radionuclides have the longest particle path length of up to several millimeters and a low LET due to the low mass of the electron (0.2 keV/μm) [91,92]. These characteristics are advantageous in medium to large tumors with heterogeneous target expression, as the β^–^-particles can reach and damage tumor cells not directly targeted by the radiopharmaceutical. This leads to an even distribution of damage and cytocidal impact. Therefore, this phenomenon was termed crossfire-effect. The major drawback of the long tissue penetration is the irradiation and possible damage to surrounding healthy tissue, which can be problematic especially in critically localized sites such as the brain or bone marrow.

In comparison, α-particles with their rather short path length (40–100 µm) and high LET are ideally suited to destroy small and critically localized tumors and metastases. They exert their cytocidal effects directly on the tumor sites, while sparing the surrounding healthy tissue. A recent case study, however, showed that α-particles can also have significant impact on large tumors [16]. As only a single patient is presented here, it is necessary to reproduce these treatment results in larger patient cohorts. The next step then would be to carefully re-assess the concept of TAT and its suitability for treating solely metastatic diseases, critically located tumors, or small tumors. Comparing the cytocidal efficiency of α- and β^–^-particles, α-particles are much more potent. This can be ascribed to their dense ionizing track structure and high RBE—a single hit of an α-particle in the cell nucleus can result in deleterious DNA double-strand breaks (DSB), leading to apoptosis induction [89,90]. At the same time, several thousands of β^–^-particles would be needed to kill a cell. Furthermore, unlike β^–^-particles, α-particles are not dependent on dose rate, active cell proliferation, or oxygenation in order to be cytocidal, making it possible to address dormant and hypoxic cells [93].

## 6. Exploiting the Indirect Radiation Effects for Tumor Control

The tumor response and ideally tumor control following endoradiotherapy is not determined solely by the direct radiation effects because there are a number of indirect effects that can be attributed to the applied radiation as well. One of these is the radiation-induced bystander effect (RIBE) [16]—irradiated cells can mediate damage signals to neighboring cells that were not irradiated. These in response can then present a similar phenotype to cells that were irradiated and ultimately also die. The exact mechanisms underlying RIBE are not fully understood but cell–cell signaling and changes in the intercellular matrix as well as extracellular reactive oxygen species might lead to changes in non-irradiated cells [15,16,90]. Another of such indirect effects of irradiation is the abscopal effect for (endo)radiotherapy. This describes the phenomenon by which localized radiation leads to immune-mediated tumor regression, not only in the irradiated area but also in regions that are locally distinct from the radiation field [16,88,90,94,95]. The underlying mechanism is most likely connected to the release of antigens and cytokines after irradiation. In turn, this leads to the stimulation of the adaptive immune system and to the selective clearance of distant tumor sites and metastases. From a clinical perspective, the immune response can be further augmented by applying systemic immune-stimulating agents as combination therapy [96].

As a pioneer study analyzing the immune response in TAT, the preclinical study of Gorin and colleagues [97,98] has to be highlighted. They assessed the ^213^Bi-induced immune modulation in a murine model of melanoma. The triggered immune response and hence the abscopal effect were mediated via cytotoxic T cells. Furthermore, danger-associated molecular patterns (DAMPs) such as Hsp70 (heat shock protein 70) and HMGB1 (high mobility group box 1) were released from treated cells and triggered the activation of dendritic cells. Additionally, Gorin et al. showed that the immune factors leading to the abscopal effect can be used for tumor-vaccination in treatment-naïve animals. Following a vaccination with plasma from irradiated animals, immune cell-mediated tumor cell death was induced in the recipient mice. A second study using the so-called diffusing α-emitters radiation therapy (DaRT)—a ^224^Ra-based brachytherapy in solid tumors of breast and colon—led to a reduction of metastases and augmented immunological memory against tumor cells [99].

In clinical trials, the combination of radiation therapy with immune checkpoint inhibitors and immunomodulators is recently of growing interest. This can be attributed to the promising results, which checkpoint inhibitors could achieve in clinical applications during tumor therapy. Therefore they are becoming an even more important pillar in cancer treatment [100,101]. The first clinical studies combining radiotherapy with ipilimumab (anti-CTLA-4-antibody; cytotoxic T-lymphocyte-associated protein 4) against prostate cancer and case reports in melanoma underline the significance of this clinical approach [102,103,104,105]. Comparably, a combination of TAT and immunomodulation might lead to specific, systemic, and durable anti-tumor response.

## 7. Atomic Nanogenerators in Modern Cancer Management

High LET radiation, as represented by atomic in vivo nanogenerators, is recommended for the treatment of radioresistant tumors [106] due to its advantageous radiobiological characteristics such as the independence of cell cycle phase and normoxia. Several pre-clinical and also clinical trials gave rise to the hope that TAT might be the “magic bullet” solution to overcome any treatment resistances [32,107,108,109,110]. As an example, we refer to one of the first reports published by Kratochwil et al. [31] using [^225^Ac]Ac-PSMA-617 as salvage therapy in mCRPC. The novel treatment resulted in an impressive treatment response for a patient presenting with progressive mCRPC after extensive pre-treatment including β^–^-endoradiotherapy with [^177^Lu]Lu-PSMA-617. However, with the growing numbers of patients treated with TAT, the occurrence of non-responders or individuals who will relapse shortly after the treatment can be observed (Figure 6). Targeted next-generation sequencing (tNGS) in responding vs. non- or only poor responding patients despite PSMA positivity to [^225^Ac]Ac-PSMA-617 therapy revealed that DNA damage-repair and checkpoint genes are frequently mutated in poor and non-responders [111]. The mechanisms as to how the inherent or acquired resistances and gene mutations develop are far from being understood, and no strategies for risk stratification of patients exist yet. 

Another major concern, especially for atomic in vivo nanogenerators, is the internalization and energy deposition of α-particles and their daughter radionuclides in healthy tissue [15,112]. Late side effects such as organ failure due to degenerative changes in the respiratory tract, kidneys, liver, bone, or gastrointestinal tract might occur. Thus far, TAT is predominantly used in the clinical setting as third-line therapy. This means the patients being treated with atomic in vivo nanogenerators had extensive pre-treatments resulting in acute and late side effects as well. Hence, to our knowledge, there are no reliable studies assessing the real extent of long-term side effects of α-emitters in human application. Dose-limiting toxicities, however, are usually well described. For [^225^Ac]Ac-PSMA-617, for instance, the destruction of the salivary glands represents the dose-limiting organ [34], whereas for [^223^Ra]RaCl_2_ haematotoxicity can lead to premature termination of the therapy [113]. Furthermore, the high LET radiation might lead to cancer induction and genetic diseases [112,114,115]. A study undertaken by Allen et al. [116], however, leads to the assumption that the clinically used doses of [^213^Bi]Bi-cDTPA-9.2.27 (cyclic anhydride of diethylenetriaminepentaacetic acid, cDTPA) do not increase the probability of secondary cancer formation in patients. After 4 weeks, even after application of a 24-fold higher dose in mice, mutation frequencies were lowered to a level close to that of spontaneous mutations.

## 8. Outlook

Targeted alpha therapies have great potential in being integrated as one of the major treatment pillars in oncology due to their astonishing treatment successes. Nevertheless, this modality has to overcome some obstacles and hurdles in order to be fully accepted. In this regard, different questions have to be carefully answered to achieve the best possible treatment option for each individual patient.

One of the major problems in TAT is the limited availability and the high costs of the radionuclides for clinical application, of which ^225^Ac in particular is to be highlighted. Currently, the available annual activity from the decay of ^233^U is about 63 GBq (1.7 Ci), which would be sufficient for less than 1000 patients per year [60,78]. This supply shortage will be most probably solved in the near future with the establishment of alternative production routes (e.g., cyclotron and reactor production) and purification strategies. On a daily basis, the work with α-emitters is regulated in a strict manner. The national and international atomic energy commissions and the individual legislations allow for significantly lower removable contamination levels, as is the case for β^–^-emitters. These restrictions further limit the handling of α-emitters and demand an advanced monitoring.

For minimizing collateral damage to healthy tissue when using atomic in vivo nanogenerators, the recoiling daughter radionuclides have to be managed efficiently and an unspecific accumulation has to be minimized. For this, stable targeting systems with a high molar activity and radiochemical yield are major milestones as well as co-treatment, for instance, with antidiuretics or free chelators. The pharmacokinetics and dosimetry of the respective radiolabeled vectors have to be studied and a major focus has to be put on cellular microdosimetry in order to achieve a maximal cytocidal effect on malignant cells. Carefully undertaken comparative pre-clinical studies might lead to the identification of the optimal radioisotopes for a tumor entity, the optimal dosing regimens (dose, fractionation, and sequence) and therapeutic strategies. Additionally, to achieve new, ground-breaking improvements in cancer care, the molecular mechanisms underlying TAT, the possible radiation resistance mechanisms, and the late-side effects should be further studied.

Answering these questions is especially important when trying to establish combination therapies with chemo- or immunotherapies in order to minimize overlapping toxic effects and yield a maximal synergistic response. Combination therapies are—unlike in other radiation therapies—not (routinely) used for TAT thus far. Therefore, the mechanisms involved in immune modulation and a presumptive role in inherent or adaptive resistance during tumor treatment need to be unraveled in more depth. Furthermore, different co-treatment strategies have to be established in order to achieve synergistic effects using combined immune and radiation therapies.

Another important factor for individualized therapies is the risk stratification of patients and the search for biomarkers and mechanisms determining inherent resistances or the rise of acquired resistances and the non-responsiveness to, among others, TAT.

The use of cocktail approaches as suggested by Haberkorn et al. [16]—applying the same or different targeting vectors with α- and/or β^–^-emitters at the same time, respectively—would exploit the complimentary nature of their particle range and their individual physical and biological characteristics. According to the initial study in patients with neuroendocrine tumors, combined endoradiotherapy with [^90^Y]Y-DOTATATE against larger lesions (high β^–^-energy), and [^177^Lu]Lu-DOTATATE against smaller lesions (lower β^–^-energy) resulted in longer overall survival than a single treatment with [^90^Y]Y-DOTATATE [117]. Combining different forms of α-particle-based therapies such as radioimmunotherapy (RIT)—which relies on longer circulating antibodies—and peptide receptor radionuclide therapy (PRRT)—with relatively fast pharmacokinetics—could potentially improve the treatment efficiency while minimizing side effects. Another interesting partner for combination with TAT could be DaRT, ^224^Ra-based brachytherapy [118,119]. DaRT was shown to be of great potential in a localized tumor treatment in both pre-clinical and clinical studies, such as in breast, colon, skin, and squamous head and neck cancer [120,121,122]. Combining this approach with TAT could lead to a synergetic effect, as well as to the selective destruction of tumors and their metastases while minimizing side effects. Moreover, another promising approach could also be the use of mixed atomic in vivo β^–^/α-nanogenerators where the mother radionuclide is represented by a β^–^-emitter. Apart from ^212^Pb/^212^Bi application [123,124], radiolanthanides such as ^147^Nd might be of interest. However, as to our knowledge, the majority of these approaches have not been tested yet. Ideally, by combining different treatment options, and carefully selecting the vectors, radionuclides, and doses, one could exploit the advantages offered by the individual substances, while keeping the side effects at a minimum.

On the basis of our knowledge and experiences with TAT, we are expecting a bright future for these approaches, especially for the ones based on atomic in vivo nanogenerators. Nonetheless, scientists and physicians have to keep working side-by-side in interdisciplinary, translational research to unravel the above discussed remaining and upcoming new obstacles, as well as to transfer their knowledge from bench to bedside and vice versa. TAT harbors the potential to be implemented as a first- and second-line treatment option and to become one of the major, fully accepted strategies in modern patient-tailored cancer management.

## Figures and Tables

**Figure 1 pharmaceuticals-13-00076-f001:**
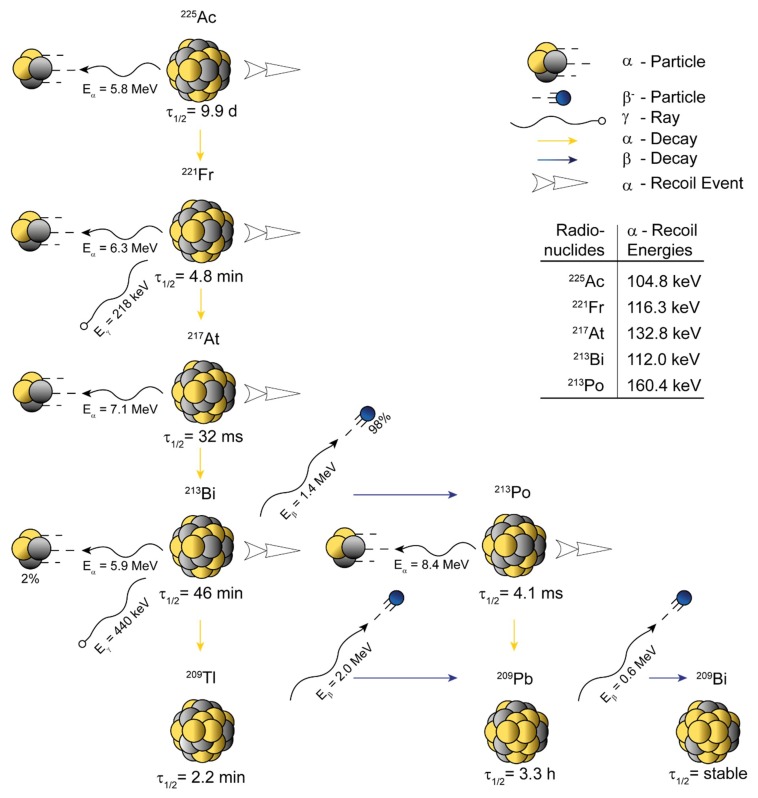
Schematic representation of the atomic in vivo nanogenerator ^225^Ac (τ_½_ = 9.9 d, E_α_ = 5.8 MeV). ^225^Ac decays through four net α-disintegrations (five in total) and two net β^–^-disintegrations (three in total) into stable ^209^Bi. The ^225^Ac decay chain possess two eligible γ-emissions for detection, 218 keV (I = 11.4%, ^221^Fr) and 440 keV (I = 25.9%, ^213^Bi). The most prominent daughter radionuclide is ^213^Bi (τ_½_ = 45.6 min, E_α_ = 5.9 MeV), which is also used for targeted alpha therapy (TAT) itself. The half-life (τ_½_), known energies connected to recoil events (translational kinetic energy E_t_), and the decay energies (E_α_, E_β_, E_γ_) are indicated on the scheme. Data were derived from Nucleonica GmbH, Nuclide Datasheets, Nucleonica Nuclear Science Portal (www.nucleonica.com), Version 3.0.65, Karlsruhe (2017).

**Figure 2 pharmaceuticals-13-00076-f002:**
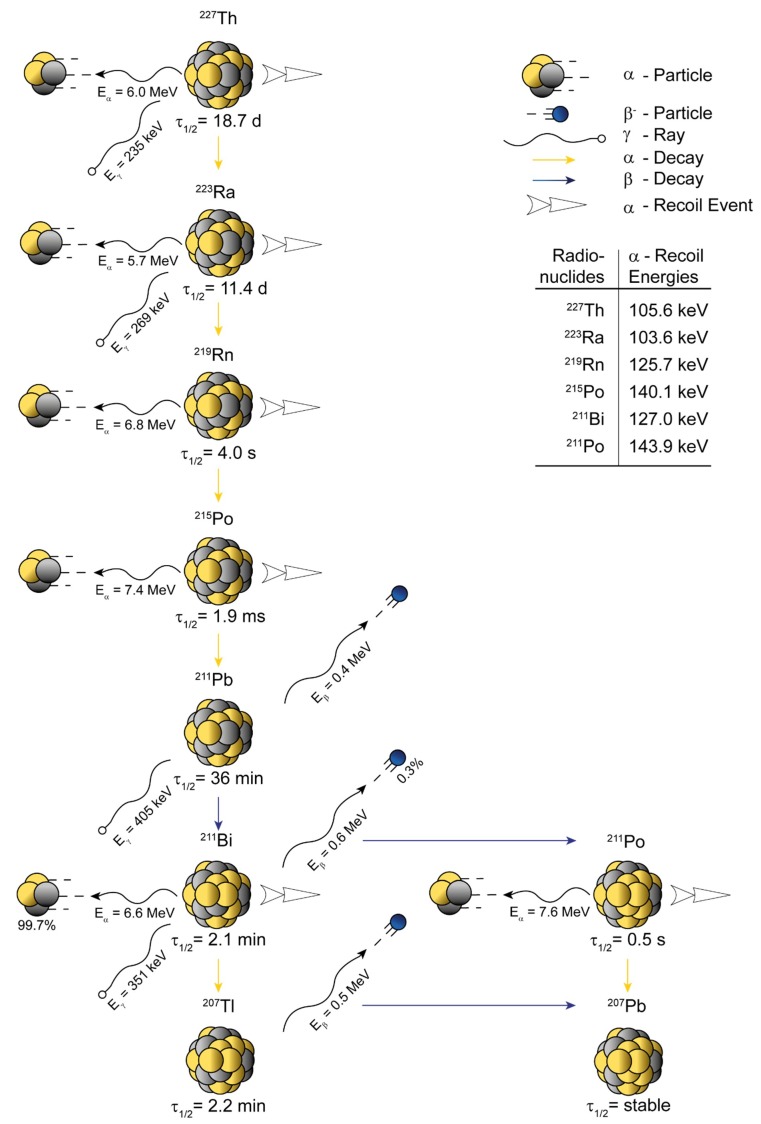
Schematic representation of the atomic in vivo nanogenerator ^227^Th (τ_½_ = 18.7 d, E_α_ = 6.0 MeV). ^227^Th decays through five net α-disintegrations (six in total) and two net β^–^-disintegrations (three in total) into stable ^207^Pb. ^227^Th possess at least four eligible γ-emissions for detection, 235 keV (I = 12.9%, ^227^Th), 269 keV (I = 13.9%, ^223^Ra), 405 keV (I = 3.8%, ^211^Pb), and 351 keV (I = 13.0%, ^211^Bi). The most prominent daughter radionuclide is ^223^Ra (τ_½_ = 11.4 d, E_α_ = 6.3 MeV), which is also used for TAT itself. The half-life (τ_½_), known energies connected to recoil events (translational kinetic energy E_t_), and the decay energies (E_α_, E_β_, E_γ_) are indicated on the scheme. Data were derived from Nucleonica GmbH, Nuclide Datasheets, Nucleonica Nuclear Science Portal (www.nucleonica.com), Version 3.0.65, Karlsruhe (2017).

**Figure 3 pharmaceuticals-13-00076-f003:**
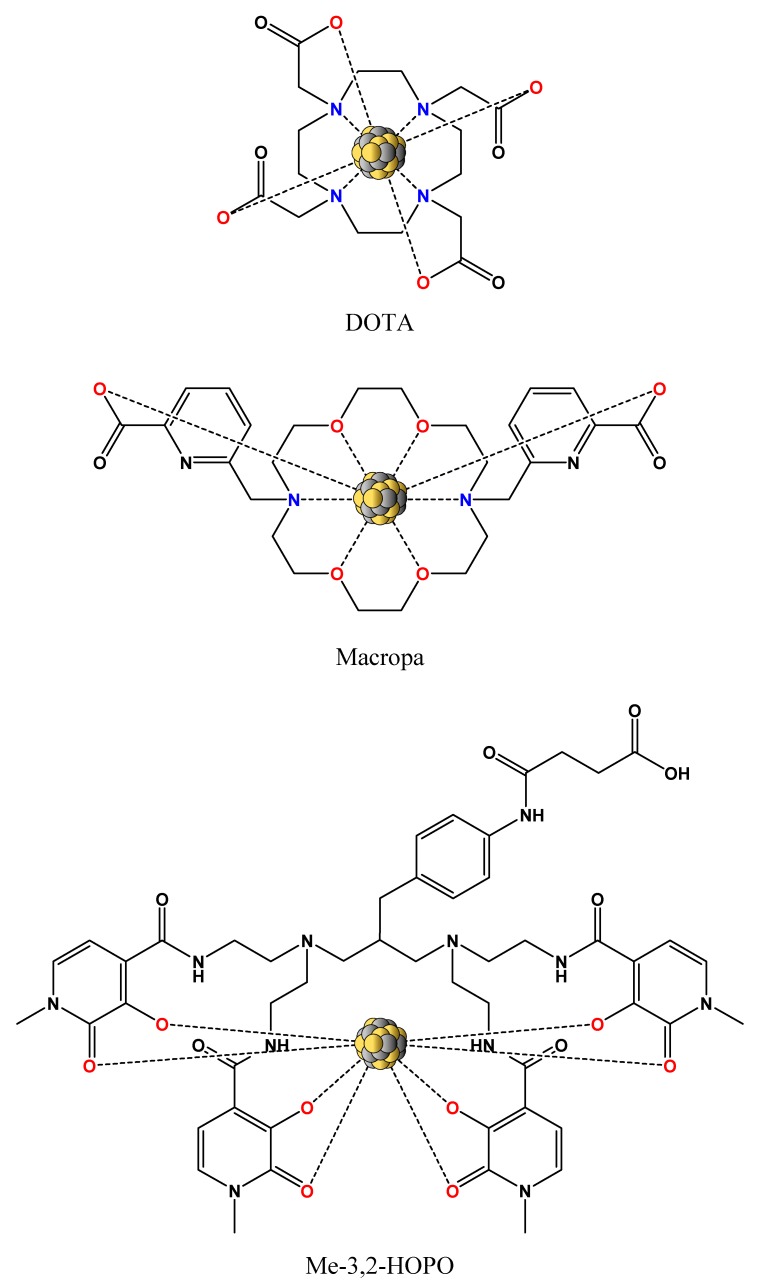
Structure of the current state-of-the-art chelating agent DOTA, next to the structures of the promising chelating agents Macropa for ^225^Ac coordination and Me-3,2-HOPO for ^227^Th coordination. Each of these chelators offers broad derivatization possibilities to further alter conjugation, labeling efficiency, and complex stability. Coordinating nitrogen donors are highlighted in blue, and oxygen donors in red. DOTA: 1,4,7,10-tetraazacyclododecane-1,4,7,10-tetraacetic acid.

**Figure 4 pharmaceuticals-13-00076-f004:**
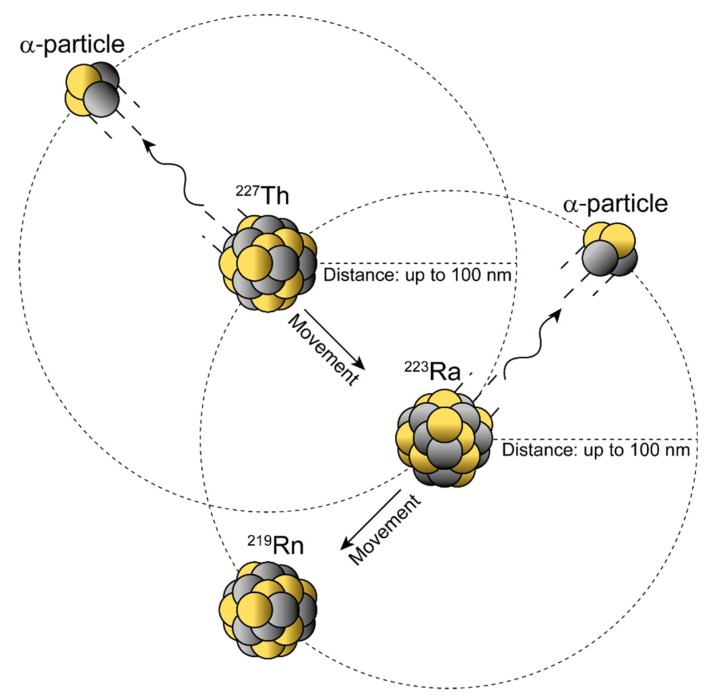
Nuclear recoil effect during α-decay within atomic in vivo nanogenerators. Schematic representation of the conservation of momentum law describing the transfer of the decay energy to the α-particle and daughter nucleus. The recoiling daughter nucleus can move up to ≈100 nm with each single α-decay and, hence, could travel considerable distances in a water-like environment (e.g., tissue). *Note*: The position of the α-particle is to show direction and not position.

**Figure 5 pharmaceuticals-13-00076-f005:**
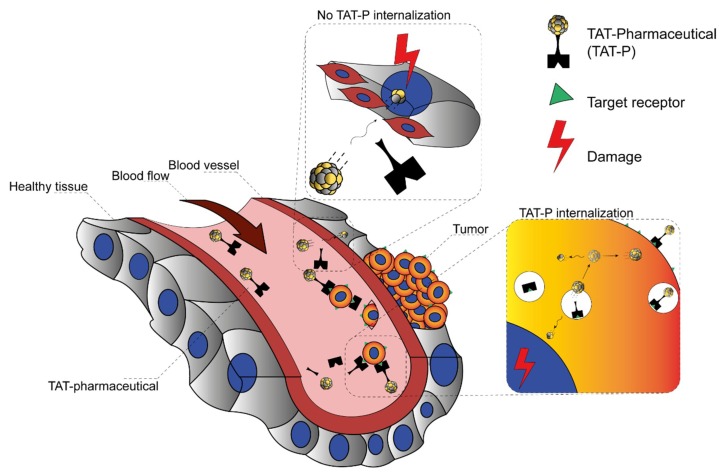
Nuclear recoil effect during α-decay within pharmaceuticals for targeted alpha therapy (TAT-P). Schematic representation of two hypothetical scenarios describing the fate of the recoiling daughter radionuclide that gets released from the chelating moiety of TAT-P in vivo. The upper section labelled “No TAT-P internalization” depicts a daughter radionuclide that is released into the blood stream while causing either unspecific local damage to healthy tissue or travels further with the blood stream and causes analogical damage distantly elsewhere. The lower section labelled “TAT-P internalization” depicts TAT-P that specifically internalizes into the targeted tumor cell. The daughter radionuclide is then released with a high probability inside the tumor cell or to a minor extent might escape the tumor cell and cause damage not only to the target tumor cell but, depending on the travelled distance, to other cells as well.

**Figure 6 pharmaceuticals-13-00076-f006:**
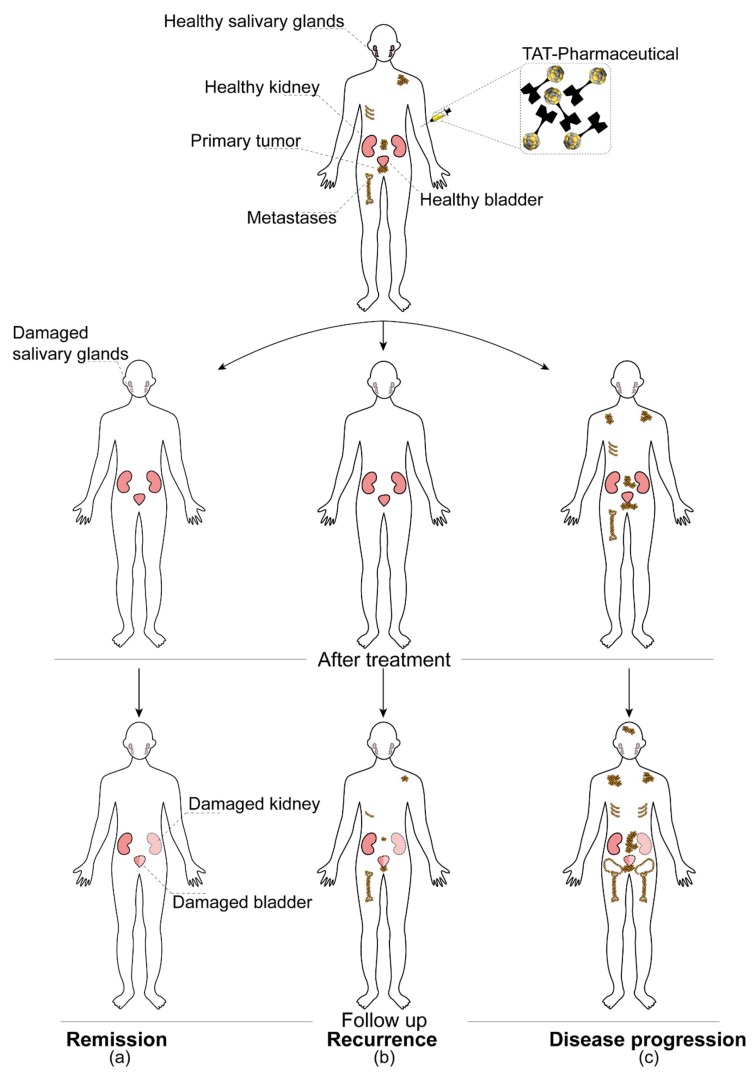
Possible outcomes of targeted alpha therapies. Prostate-specific membrane antigen (PSMA)-targeted alpha therapy of metastasized castration-resistant prostate cancer (mCRPC) was selected as an example for the schematic representation of three main hypothetical scenarios. These are (**a**) full and long-lasting remission, (**b**) disease recurrence after the successful outcome of the initial therapy, and (**c**) disease progression, which occurs in spite of the applied therapy (therapy resistance). In addition, possible early and late side effects, such as salivary gland, kidney, and bladder damage, are also indicated in the figure. TAT is usually applied as a third-line or salvage therapy when all other previous treatment options have failed.

**Table 1 pharmaceuticals-13-00076-t001:** Selection of various characteristics demonstrating the striking differences between ^225^Ac and its first three progenitors in relation to their diverse coordination preferences.

	^225^Ac	^221^Fr	^217^At	^213^Bi
Character	Actinide	Alkali metal	Metalloid	Post-transition metal
Oxidation state *	+3	+1	−1, +1 & +7	+3
Effective ionic radius [pm] ^#^	112 (6)	180 (6)	62 (6) ^$^	103 (6)

* Most common oxidation state; ^#^ Number in brackets determines the coordination number; ^$^ Effective ionic radius for oxidation state +7; Data for effective ionic radii were derived from http://abulafia.mt.ic.ac.uk/shannon/ptable.php.

**Table 2 pharmaceuticals-13-00076-t002:** Selection of various characteristics demonstrating the striking differences between ^227^Th and its first three progenitors in relation to their diverse coordination preferences.

	^227^Th	^223^Ra	^219^Rn	^215^Po
Character	Actinide	Alkaline earth metal	Noble gas	Post-transition metal
Oxidation state *	+4	+2	0	+2 & +4
Effective ionic radius [pm] ^#^	94 (6)	148 (8)	n.d.	94 (6) ^$^

* Most common oxidation state; ^#^ Number in brackets determines the coordination number; ^$^ Effective ionic radius for oxidation state +4; n.d. = not determined; Data for effective ionic radii were derived from http://abulafia.mt.ic.ac.uk/shannon/ptable.php.

**Table 3 pharmaceuticals-13-00076-t003:** Selection of different (radio)nuclides related to atomic in vivo nanogenerators ^225^Ac and ^227^Th and their affinity to various organs.

Element	Radioisotope	Targeted Organ	Reference
Actinium	^225^Ac	liver, bone, kidney	[62]
Francium	^221^Fr	kidney	[63]
Bismuth	^211^Bi and ^213^Bi	kidney	[64,65]
Lead	^209^Pb and ^211^Pb	liver, bone, kidney	[66]
Polonium	^211^Po * and ^213^Po	liver, kidney, red bone marrow, spleen	[67]
Thorium	^227^Th	liver, bone, spleen	[68]
Radium	^223^Ra	bone, soft tissue, intestine	[69,70,71]
Astatine	^217^At ^$^	thyroid, stomach, lung	[72,73]
Thallium	^207^Tl and ^209^Tl ^#^	heart, kidney, large bowel, thyroid, testicles	[74]

* Probability of only ≈0.3% in the decay chain of ^227^Th; ^$^ With a half-life of only 32 msec, the accumulation of ^217^At could not be proven in these organs but is assumed to be similar to that of other astatine isotopes; ^#^ Probability of only ≈2% in the decay chain of ^225^Ac.

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
