# Peer review of "Atomic Nanogenerators in Targeted Alpha Therapies: Curie’s Legacy in Modern Cancer Management"

_pharmaceuticals, 2020, doi:10.3390/ph13040076_

Round 1
Reviewer 1 Report
This manuscript is a review and perspective of the state of targeted alpha therapies for cancer therapy. I spent quite a bit of time debating with myself the interpretation of the question in the title. It does create intrigue, but the manuscript only concludes with maybe. In spite of the relative scarcity of these alpha-emitting radionuclides, significant work has been conducted to suggest the utility. Alpha-emitting radiotherapeutics seem to have two major challenges; 1) the need to get high uptake in target quickly to minimize alpha particle damage elsewhere (i.e. in the circulation) and 2) the release of daughter alpha emitters that tend to accumulate in other tissues and cause damage (i.e. kidney or bone). The number of reviews on alpha therapy seems proportionally high.
The manuscript was an enjoyable read until the section entitled Radiobiology. The first reason may be that the biology is my personal weakness, however, the organization of this section may need more thought. The second paragraph (following a one sentence paragraph) has 28 lines and two major topics. I suggest separating, into at least 2 paragraphs, the discussion of beta versus alpha particle for effective tumor killing from the discussion of RIBE, abscopal, and other immunology components.
The desire to limit migration of daughter nuclides, relevant for Ac-225 more so than Pb-212, by use of nanoformulations seems important. However, the review by Robertson et al. from TRIUMF (2018) included only 6 references to nanoformulation approaches to contain daughter nuclides. The latest of these was published in 2016. The standard nanoparticle approach papers suggest that the outer envelope of the particle needs to be stronger (thicker) to contain the recoil nuclide, but will this strategy also attenuate the alpha particles?
Figures. I appreciate your work on the figures. They are clearly depicted. I hope editor does not complain about the size.
Titles in references have proper nouns that need capitalization. Curie, von Hevesy, etc.
Line 106 – In contrast to Ac-225, or Contrary to Ac-225.
Figure 1 – is Et = 1132.8 correct for Bi-213? I do not know how these are calculated.
Figure 4. The alpha particle position is not to scale. It would not be practical to draw to scale. Perhaps it would be best to merely state in legend that position of alpha particle is to show direction not position. The velocity of the recoil is about 2% of the velocity of the alpha particle. How do you calculate the distance traveled by the daughter? There must be some factor for density of the medium. This is discussed in reference 34 which is also a review published in 2018.
Reference 34 is incomplete. No page number is given. Kozempel 2018, 23(3), 581.
Author Response
Please, see the attachment.

Reviewer 2 Report
The paper ‘Atomic Nanogenerators in Targeted...’ by Roscher et al reviews the challenges of using alpha emitting radionuclides such as Ac-225, Th-227 and Ra-223 which decay through several decay steps before they reach a stable isotope. As much as I liked the paper my main concern is what new information does this paper provide in comparison to previous reviews. In addition, the review appear rather incomplete, it mentions examples but it does not properly cover the available literature and does not even elaborate on the provided examples. That is acceptable if the review is a critically review but that it has currently not enough depth to be considered a critical review.
I have the following minor comments:
- Why are the decay series called nanogenerators? That implies certain size. Why nano? It has certainly nothing to do with the size of the radioisotopes.
- On page 1 line 37, X-rays are not a discipline. The same sentence is too long and totally not understandable
- Page 2 line 44, provide the original reference, rather than cite a review.
- Page 2, line 53. Explain why having 2+ charge is favourable for therapy, this is not clear at all at the moment.
- Page 2, line 59, the word ‘within’ in this context is not correct.
- Page 2, line 64, a reference should be added.
- How is Radium mimicking the structure of Ca? In what way? Same oxidation state, chemical reactivity.... That should be explained.
- Page 2, line 74, is relapse a side effect? Or is it due to not full elimination of cancer cells?
- Page 3, the chapter title refers to physico-chemical properties but is nuclear decay a pshysico-chemical property?
- Page 4, line 103. What do the authors mean by 12 coordination numbers?
- Page 10, line 210. What do the authors understand under biological uptake, uptake in cells or in organs or..?
- In the same chapter, consider the time that a radionuclide travels to the target cells, even if there is fast uptake there might be significant release of recoils. A simple calculation can help to evaluate how big the problem is and will give some depth to the review.
- The biological affinity of Fr-221 and At-217 might not be know but for other isotopes of the same elements the biological affinity is known and should be mentioned.
- I think that an interesting point that the review makes is to use chelators or amino acids to reduce side effects caused by the recoiling daughters. This part is interesting and should be extended, now it is just a small paragraph.
- The radiobiology should focus more on the effects of alpha’s. The bystander effect is a typical example which occurs not only with alpha’s. So why is it mentioned?
- In the radiobiology parts it is mentioned that alpha’s can also have impact on large tumours. More information should be provided, what impact, size reduction, elimination etc? And how is this explained.
- It also looks like the radiobiology goes over into a conclusion part (from line 305 on). It is better to have clear conclusions without mixing with the main part.
Author Response
Please, see the attachment.

Reviewer 3 Report
Review of Roscher et al. – Atomic Nanogenerators in Targeted Alpha Therapies: Can broader knowledge of natural phenomena defeat the ignorant beast in this radiant fairy tale of science?
Article summary:
Roscher et al. present a review article which aims to provide insights into the challenges and limitations of radionuclide therapy with ‘alpha generator’ radionuclides (radionuclides with progeny decaying via alpha emission), and how these challenges could potentially be overcome by rational optimization strategies based on better understanding of underlying physical, biochemical, and radiobiological principles. It is likely to be a useful resource for newcomers to the field of targeted alpha (generator) therapy (TAT).
General comments:
The review in its present form is somewhat similar to others (e.g. Poty et al.; doi: 10.2967/jnumed.116.186338 and doi: 10.2967/jnumed.117.204651) and is thus not particularly novel – however, I don’t think that should preclude it from publication in Pharmaceuticals, as several aspects are approached from distinct angles, and it is useful to the field to have multiple perspectives on this subject.
Specific comments:
TITLE, ABSTRACT
The verbiage “radiant fairy tale of science” in the title and abstract, suggest to the reader that alpha therapy is some pie-in-the-sky concept of science fiction, when it most certainly isn’t. Similarly, for the “ignorant beast” verbiage – the challenges here (and many solutions thereof) are well-known to experts in TAT, and in my opinion, it is preferable, for the title and abstract, to find alternate wording to express the aim to highlight/underscore these considerations for their educational value.
However, the quote from Marie Curie in the introduction is a very thoughtful touch – I think it’s entirely appropriate to make the analogy of alpha therapy as a ‘fairy tale’ in the introduction following the quote – just not in the title/abstract where the context is absent and the authors’ meaning is likely to be misinterpreted.
- INTRODUCTION
Line 48: Please use the recently published Consensus Nomenclature Rules (doi: 10.1016/j.nucmedbio.2017.09.004) for naming of radioactive compounds throughout the manuscript (e.g. [223Ra]RaCl2 as opposed to 223RaCl2).
Line 79: Perhaps revise stating the intention is “… connecting examples of the phenomenological aspects [e.g. radiobiological, etc. alluded to in the preceding sentence], with the various scientific achievements and clinical successes … “ – since the aim of the article is to advance understanding of the relationships between the successful application of TAT and the underlying science.
Line 81: “define new individualized treatment schemes” – either expound on your meaning here or remove this phrase, as definition of new individualized treatment schemes is not addressed in this review (nor should it be).
- PHYSICOCHEMICAL PROPERTIES OF Ac-225 AND Th-227
Line 88: “which limits the use of readily available optical spectroscopic techniques for actinium detection” Suggest to remove – such methods are not routinely used for detection of alpha emitters (only for probing the structure of in specialized settings like in the referenced article); moreover these are nuclear spectroscopic methods, not optical.
Line 103: Thorium can possess coordination number of up to 12 (this is distinct from saying thorium exists stably in 12 different coordination states, which is implied by the wording)
- COORDINATION CHEMISTRY
Line 160: “This complex should be of utmost importance …” should be clarified for educational value – e.g. for chelation of Fr-221 released following recoil in vivo (potentially important)? For chelation of Fr-221 in radiopharmaceutical formulation (likely not important)?
- NANOGENERATORS AND THE NUCLEAR RECOIL EFFECT
Line 188-190: “For example, Ac-225 was shown to be approximately 1,000 times more potent in its cytocidal effect than its daughter Bi-213 alone. This difference in the potential of both radionuclides is also intensified by the fact that Ac-225 has >300-fold longer half-life than Bi-213” This needs clarification: the 1000-fold increase in potency is on a per-unit-administered-activity basis and thus includes the effect of longer half-life. Without the additional half-life, it would be ~4 times as potent (since it emits 4 net alphas where Bi-213 emits just one).
Line 244: “The in vivo proof-of-concept, however, is still missing …” It has already been shown that nanoparticle daughter retention works in vivo (McLaughlin et al - doi: 10.1371/journal.pone.0054531)
- OUTLOOK
Line 352: Need reference for figures on availability. Also a typographical error – 63 GBq is 1.7 Ci (not mCi)
FIGURE 1 –
Need source for decay energy data referenced. In particular, the recoil energies shown appear highly suspect (should be around 100 keV). Why is the recoil energy not provided for the parent?
Line 96: Replace dominant with net. Ac-225 emits four net alpha emissions. Two net beta emissions.
FIGURE 2 –
Need source for data referenced, as above. Th-227 emits five net alphas, two net betas.
Author Response
Please, see the attachment.

Reviewer 4 Report
Thank you, authors, for the attempt to cover a large and quickly developing field of targeted alpha therapy delivery methods. The issues with in vivo complex stability and the fate of the recoiling daughter radionuclides are indeed substantial problems to be solved yet. But despite the pleasure of reading poetically metaphoric sentences, there are several flaws to be addressed. Please see in more detail below.
Line 44. The term targeted alpha therapy was introduced in the late 90s.
Line 59. Authors state that Actinium 225 half-life is 9.9 days, but late in a text ( Line 89) for example) it stated as 10.0. Please provide the correct value for the half-hile of Actinium 225.
Line 75. Please, provide the reference for the statement: "off-target-effects caused by the nuclear recoil effect".
Line 91. Actinium 225 has 121+ isomeric gamma lines due to the decay. At least several have about 1% abundance per decay. It can be registered and measured by gamma spectroscopy detectors. Please see references: [McDevitt M.R. at al, Science. 2001; Tichacek C.J. et al, Molecules. 2019; White M.C. at al, Photoatomic Data Library MCPLIB04: A New Photoatomic Library Based on Data from ENDF/B-VI Release 8. Los Alamos National Laboratory; Los Alamos, NM, USA: 2002.]
Line 197. Please provide more information on the classical approach to calculate recoil energy. Do you have references on experimental data on the recoil effect for alpha emitters? Please provide them also.
Line 275. Please correct the statement “low let is due to small charge of electron ”. See references : [Bethe H. Bremsformel für Elektronen relativistischer Geschwindigkeit. Zeitschrift für Physik. 1932;76:293–299 ; Grimes D.R., Warren D.R., Sci Rep. 2017;7(1); ]
Lines 279,283,295. Please, define the difference between abscopal and bystander effect more clearly.
Author Response
Please, see the attachment.

Reviewer 5 Report
This review by Roscher et al. presents a brief summary of the obstacles associated with the use of atomic nanogenerators (Ac-225, Th-227, Ra-223/224) for cancer treatment. The authors encompassed different areas related with radionuclides (physicochemical properties, coordination chemistry), nuclear recoil effects, and radiobiology, aiming to highlight the areas that required additional work (e.g., “ignorant beast”) to advance targeted alpha therapy. The need for a comprehensive understanding of the nuclear and biological phenomena associated with targeted alpha therapy is critical to define this therapy as a first or second line of cancer treatment. I recommend this review for publication in Pharmaceuticals after the authors addressed the following points.
Please consider the following points:
- As a review, a more comprehensive literature review evaluation must be done in the different areas being covered. For example:
o Include references for different targeting vectors (aptamers, monoclonal antibodies, peptides), chelating agents, and nanocarriers used in combination with nanogenerators. For example, there has been extensive work done in developing nanoparticles to minimize the relocation of decay daughters and the authors only referenced the work done by Piotrowska et al. (2017). Please consider the work done with polymersomes, liposomes, lanthanide phosphate, lanthanide vandates, titanium dioxide, superparamagnetic iron oxide, among others. These references should complement the sentences in line 64 and lines 242-250. In addition, expand on the different chelators tested and used for Ac-225, Ra-223, and Th-227.
o Include references that described the work that has been done in terms of radiobiology. For instance, in lines 275-276 reference literature showing that DNA DSBs lead to apoptosis. Pouget et al. (2015) presented key aspects regarding the radiobiology of targeted radionuclide therapies and the difference with external radiationn therapy.
- Line 90-91. Clarify the term “well detectable” regarding the emission of gamma rays from Ac-225. It is clear that the low intensity of gamma rays from Ac-225 makes their detection challenging, however, it is possible to detect them using gamma spectroscopy.
- The sentence in line 163-164 seems to be out of context. Please give context of why it is important or not that the labeling of these ligands is performed at room temperature for few minutes. Same idea applies for sentence in lines 171-172.
- Line 188. Provide more details regarding the higher levels of cytotoxic radiation in target tissues of alpha particles. Is this comparison being made with respect to beta or external beam?
- Include a reference for the accumulation of Ac-225 in bone and liver.
- Nanoparticle biodistribution does not only depend on the EPR effect (passive targeting). Nanoparticles can be functionalized and conjugated with different targeting vectors for active targeting of tumor tissue. Include sentences that clarify this in your discussion.
- Why there is no physicochemical description for Ra-223/224 in this review? As mentioned by the authors, Xofigo has motivated the interest in developing novel radiopharmaceuticals based on alpha emitters.
- Line 188-189, reference for the greater cytotoxic potential of Ac-225 compared to that of Bi-213.
- Line 196, please revise the concepts of binding energy and chemical bond energy.
- Line 334, complement sentence “Dose-limiting toxicities, however, are usually well described”. Give context and include references.
- Line 352, 63 GBq = 1.7 Ci.
Editorial suggestions:
- Be consistent in the reported half-life for different isotopes. For Ac-225 a half-life of 9.9 d and 10.0 d is used in the manuscript, whereas for Th-227 it was reported 18.9 d and 18.7 d.
- On a similar idea, there is no need to repeat the half-life of the radionuclides throughout the text.
- Include the gamma ray energies for Bi-211 and Pb-211 in Fig. 2 since these decay daughters can be detected by gamma ray spectroscopy.
- Line 272, use past tense in this sentence instead of “recent studies, however, could show…”.
- No need for (NHL) acronym because it was not used again in the text.
Author Response
Please, see the attachment.

Round 2
Reviewer 2 Report
No comments. Paper has improved sufficiently to allow publication.
Author Response
The authors are grateful for the positive judgement on our manuscript!
Reviewer 3 Report
Review of Roscher et al. Revised Manucript
Overall I recommend acceptance of the article as the authors seem to have addressed most of my & other reviewers' comments. However, I found the following minor issues in the revised document should still be addressed.
Line 55-56: Suggest to omit ‘two positive charges (dense ionization due to a high stopping power)’ – this phrase is redundant as the linear energy transfer is introduced later in the sentence.
Line 271-274: The authors misinterpret the results of the referenced study – this study (ref. 79) in fact utilized a targeted approach (the nanoparticles were conjugated to antibodies for active targeting) – the lung accumulation was due to the fact that the antibody used was specific for thrombomodulin receptors in the lung tissue – not due to non-specific (EPR) accumulation.
Line 313: Typo in the units for the electron LET: should be 0.2 keV/micrometer
Fig. 1 and 2: I understand the authors’ reasoning they explained for labeling the daughter nuclides with the energies imparted to them from alpha recoil. However, this is generally inconsistent with how nuclear decay data are presented (it is almost universal practice to associate decay energies with the mother nuclide). Adding further confusion to the way the recoil energies are presented, is that they are presented paired with (they’re directly above) the arrows & energies of the daughter alpha emissions. If the authors wish to list the recoil energy above the arrows designating alpha emissions, I’d suggest to list the corresponding recoil energy there (not below/associated with the alpha decay of the daughter). Lastly, the recoil energies are missing in some cases, and mixed up in other cases (for example, in Fig. 1, 217At was skipped, and the data for 213Bi is in its place; in Fig. 2, some are swapped). See below - these should match the Nucleonica datasets within 1 or 2 keV.
Mean alpha recoil energies from ICRP Publication 107 (nuclear data for dosimetric calculations), for the 225Ac chain:
225Ac: 104.8 keV
221Fr: 116.3 keV
217At: 132.8 keV
213Bi: 112.0 keV
213Po: 160.4 keV
Mean alpha recoil energies for the 227Th chain:
227Th: 105.6 keV
223Ra: 103.6 keV
219Rn: 125.7 keV
215Po: 140.1 keV
211Bi: 127.0 keV
211Po: 143.9 keV
Author Response
Please, see the attachment.

Reviewer 5 Report
The authors have addressed all comments and suggestions successfully. I recommend this article for publication in pharmaceuticals. Please keep the hard work to advance targeted alpha therapy as treatment strategy for cancer and infectious diseases.
Author Response
The authors are grateful for the positive judgement on our manuscript and also for the supportive attitude towards targeted alpha therapy!